# Platelet Count in Patients with SARS-CoV-2 Infection: A Prognostic Factor in COVID-19

**DOI:** 10.3390/jcm11144112

**Published:** 2022-07-15

**Authors:** Andrea Boccatonda, Damiano D’Ardes, Ilaria Rossi, Alice Grignaschi, Antonella Lanotte, Francesco Cipollone, Maria Teresa Guagnano, Fabrizio Giostra

**Affiliations:** 1Emergency Department, IRCCS Azienda Ospedaliero, Universitaria di Bologna, 40138 Bologna, Italy; andrea.boccatonda@gmail.com (A.B.); alice.grignaschi@aosp.bo.it (A.G.); antonella.lanotte@aosp.bo.it (A.L.); fabrizio.giostra@aosp.bo.it (F.G.); 2Institute of “Clinica Medica”, Department of Medicine and Aging Science, “G. D’Annunzio” University of Chieti, Vestini Road, 66100 Chieti, Italy; rossilaria91@gmail.com (I.R.); francesco.cipollone@unich.it (F.C.); guagnano@unich.it (M.T.G.)

**Keywords:** platelet, COVID-19, SARS-CoV-2, pneumonia, coagulopathy

## Abstract

COVID-19 patients may manifest thrombocytopenia and some of these patients succumb to infection due to coagulopathy. The aim of our study was to examine platelet count values in patients infected with SARS-CoV-2, comparing them to a control group consisting of non-COVID-19 patients. Moreover, we evaluated the correlation between the platelet value and the respiratory alteration parameters and the outcome (hospitalization and mortality) in COVID-19 patients. The mean platelet values (×10^9^/L) differed between patients with positive or negative SARS-CoV-2 swabs (242.1 ± 92.1 in SARS-CoV-2 negative vs. 215.2 ± 82.8 in COVID-19 patients, *p* < 0.001). In COVID-19 patients, the platelet count correlated with the A-aO_2_ gradient (*p* = 0.001, rho = −0.149), with its increase over the expected (*p* = 0.013; rho = −0.115), with the PaO_2_ values (*p* = 0.036; rho = 0.093), with the PCO_2_ values (*p* = 0.003; rho = 0.134) and with the pH values (*p* = 0.016; rho = −0.108). In COVID-19 negative patients, the platelet values correlated only with the A-aO_2_ gradient: (*p* = 0.028; rho = −0.101). Patients discharged from emergency department had a mean platelet value of 234.3 ± 68.7, those hospitalized in ordinary wards had a mean value of 204.3 ± 82.5 and in patients admitted to sub-intensive/intensive care, the mean value was 201.7 ± 75.1. In COVID-19 patients, the survivors had an average platelet value at entry to the emergency department of 220.1 ± 81.4, while that of those who died was 206.4 ± 87.7. Our data confirm that SARS-CoV-2 infection may induce thrombocytopenia, and that the reduction in platelet counts could be correlated with the main blood gas parameters and with clinical outcome; as a consequence, platelet count could be an important prognostic factor to evaluate and stratify COVID-19 patients.

## 1. Introduction

Coronavirus Disease 2019 (COVID-19), caused by the Severe Acute Respiratory Syndrome Coronavirus-2 (SARS-CoV-2), can simultaneously manifest in different organs as well as the pulmonary system [1,2]. COVID-19 patients present a wide range of clinical conditions, ranging from asymptomatic infections, minimal symptoms (the majority of patients) to fatal respiratory distress (ARDS) [1]. Data in published literature have shown that about a quarter of COVID-19 infected patients experienced thrombocytopenia and about half of these patients succumb to infection due to coagulopathy [3,4]. The mechanisms leading to thrombocytopenia are not yet well known. The aim of our study was to examine platelet count values in patients infected with SARS-CoV-2 virus and compare it to a control group of subjects with a negative swab. Moreover, we correlated the platelet value with respiratory alteration parameters and with the outcome of COVID-19 patients, attempting to explain the molecular and physio-pathological hypothesis at the basis of the observed data.

## 2. Material and Methods

A retrospective analysis was performed on patients with suspected SARS-CoV-2 infection who referred to the Emergency Department of the Sant’Orsola-Malpighi Hospital in Bologna from March 2020 to May 2020. All patients underwent a molecular nasopharyngeal swab for the diagnosis of SARS-CoV-2 infection at entrance to the Emergency Department and were subjected to arterial blood gas analysis and blood exams. The platelet reference range was 160–370 × 10^9^/L. Data collected from blood gas analysis were as follows: partial pressure of oxygen (pO_2_; mmHg), partial pressure of carbon dioxide (pCO_2_; mmHg), PaO_2_/FiO_2_ ratio, alveolar-arterial gradient (A-aO_2_ gradient). Vital parameters such as respiratory rate and peripheral oxygen saturation were also recorded.

The outcome of discharge from the emergency room was also analyzed, dividing patients into those who were discharged, hospitalized in ordinary wards and hospitalized in the sub-intensive/intensive care unit. Moreover, a differentiated analysis was performed on surviving patients and those who had died,30 days after emergency department admission.

### Statistical Analysis

Continuous variables are expressed as mean ± standard deviation. The Mann–Whitney U test for independent samples was used to compare the quantitative variables between groups. Categorical variables are presented as frequencies and percentages and compared using the Chi-squared test with Yates’ correction. The correlation between variables was measured using Pearson’s coefficient. A *p* value < 0.05 was considered statistically significant. All data were collected and entered into an Excel database (Microsoft Office 2016, Microsoft, Redmond, WA, USA) and statistical analyses were performed using SPSS (IBM SPSS Statistics 25 Version, Inc., Chicago, IL, USA).

## 3. Results

A total of 998 patients were examined. A COVID-19 positive swab was detected in 489 patients, while 509 patients had negative swab (see patients’ characteristics in Table 1). COVID-19 patients more frequently had hypertension, diabetes and COPD. Table 2 and Table 3 show patients’ symptoms and vital parameters on arrival at the emergency department. The mean platelet counts were different between the two groups (242.1 ± 92.1 in COVID-19 negative vs. 215.2 ± 82.8 in COVID-19 patients, *p* < 0.001) (see Figure 1). In COVID-19 patients, the platelet count correlated with the A-aO_2_ (*p* = 0.001, rho = −0.149) (see Figure 2), with its increase compared to the age-expected (*p* = 0.013; rho = −0.115), with the values of PaO_2_ (*p* = 0.036; rho = 0.093) (see Figure 3), with the values of PCO_2_ (*p* = 0.003; rho = 0.134) and with the pH values (*p* = 0.016; rho = −0.108), while it did not correlate with peripheral O_2_ saturation values, respiratory rate and PaO_2_/FiO_2_ ratio. Moreover, we observed that the value of platelets inversely correlated with C-reactive protein with a statistical significance (*p* < 0.001; Rho = −0.146) (Figure 4). On the contrary, platelets count did not correlate with procalcitonin (*p* = 0.96; Rho = 0.00) (Figure 5). In COVID-19 negative patients, the mean platelet count correlated only with the A-aO_2_ (*p* = 0.028; rho = −0.101), while no correlation with the other blood gas analysis parameters was detected. Regarding the outcome of the emergency department admission of patients with COVID-19, the discharged patients (*n* = 128) had a mean platelet value of 234.3 ± 68.7, those hospitalized in ordinary wards (*n* = 329) had a mean value of 204.3 ± 82.5 and those admitted to intensive/sub-intensive care (*n* = 32) had an average value of 201.7 ± 75.1. In the group of COVID-19 patients, the survivors had an average platelet value at entry of 220.1 ± 81.4, while for those who died it was 206.4 ± 87.7.

## 4. Discussion

In our study, we demonstrated how platelet count values differed between the two groups (COVID-19 and non-COVID-19 patients), with lower values in COVID-19 patients. The hypothetical mechanisms by which SARS-CoV-2 causes thrombocytopenia are the following: impaired haematopoiesis caused by systemic inflammation or cytokine storm, such as IL-6 which is frequently elevated in SARS-CoV-2 infection [5]; SARS-CoV-2 might directly infect haematopoietic stem cells or megakaryocytes through angiotensin-converting enzyme 2, CD13 or CD66a [5]; antiviral antibodies could cross-react with haematopoietic cells and/or platelets, as observed for anti-adenovirus antibodies which can cross-react with platelet integrin GPIIb/IIIa [5,6]; impaired maturation of megakaryocytes in COVID-19 patients [7]; thrombotic microangiopathy and disseminated intravascular coagulation leading to the increased consumption of platelets revealed by autopsy of non-survivors [5,8]; activated platelets scavenging by splenic/hepatic macrophages [5]. Some of the mechanisms implicated in COVID-19 pathogenesis, including hypoxia, inflammation, immune system activation and endothelial activation and dysfunction, are known to stimulate platelet activation and apoptosis, leading to increased thrombosis [9]. Furthermore, the presence of apoptotic platelets promotes hyperactivation of surviving platelets [9]. The most interesting data revealed by our analysis is that the platelet value was statistically significantly correlated with the values of pO_2_, A-aO_2_ gradient and its increase compared to the expected for age. Therefore, there is an inverse correlation between the degree severity of respiratory compensation and the platelet value. In particular, a direct correlation is also evident between the degree of compromise of gas exchanges at the alveolar-capillary membrane level, as evidenced by the A-aO_2_ gradient, and the platelet values. The fact that the value of platelets correlates with the degree of respiratory impairment could explain the finding that patients discharged to home have higher platelet values than those who are hospitalized, while patients admitted to sub-intensive or intensive care units requiring ventilatory support have the lowest values. Moreover, platelet count inversely correlated with C-reactive protein which represents an important index of systemic inflammation. On the contrary, platelet count did not correlate with procalcitonin; this aspect could be related to the specific role of procalcitonin, that represents an index of bacterial sovrainfection that is present only in some patients, especially in the case of SARS-CoV-2 infection. The platelet value detected on first medical evaluation appears to be a prognostic factor of mortality; in fact, as shown, COVID-19 patients who died within 30 days displayed lower platelet values. In published literature, several studies have demonstrated that platelets play a main role in the development of respiratory distress following lung injury. Depletion of platelets induces increased alveolar bleeding, dissemination of bacteria, plasma cytokine counts and disease severity, thus suggesting a primarily protective role for platelets [10]. Moreover, thrombocytopenia has been shown to be a risk factor for ARDS and prognostic of poor outcomes among patients with ARDS [11,12]. Platelets seem to act as immunomodulatory cells that can be generated in lungs, thus carrying several immunomodulatory proteins, and those generated in the lung may be more sensitive to bacterial and viral exposure due to increased presence of TLRs [13]. Therefore, neutrophils that bind activated platelets have a potent and diversified immune arsenal at their disposal. Furthermore, a recent study by Kanth Manne et al. demonstrated that platelets from COVID-19 patients aggregated faster and showed increased spreading on both fibrinogen and collagen [14]. The increase in platelet activation and aggregation could partially be attributed to increased MAPK pathway activation and thromboxane generation [14]. These findings demonstrate that SARS-CoV-2 infection is associated with platelet hyperreactivity, which may contribute to COVID-19 pathophysiology [14]. Moreover, platelet P-selectin is a key thromboinflammatory molecule involved in platelet activation and function and its level has been shown to correlate with acute lung injury severity score and related death [15]. Several autopsy studies in patients with COVID-19 have reported thrombosis in arteries and in veins, particularly within small vessels (platelet-rich thrombotic microangiopathy), and the presence of thrombosis throughout the lungs suggests that hyperactivated platelets might additionally influence disease in the lungs by exacerbating inflammation [16].

Unfortunately, in our work, it was not possible to perform an assessment of adenosine diphosphate (ADP)-, thrombin receptor activator peptide 6 (TRAP)-, and arachidonic acid (AA)-induced platelet activity through impedance aggregometry. That method would have allowed us to further define the prognostic impact of thrombocytopenia, but the method is not widely available in an emergency room setting. Heinz and colleagues demonstrated greater fibrinolysis resistance in COVID-19 patients by thromboelastometry, but they did not find a greater platelet aggregability based on impedance aggregometric tests [17]. A subsequent study performed on 18 patients admitted to the ICU with COVID-19 induced ARDS revealed hypercoagulability and hypoactive platelet dysfunction, but these were poorly correlated with thromboembolic or bleeding complications [18].

## 5. Conclusions

SARS-CoV-2 infection can induce different degrees of thrombocytopenia depending on the severity of the disease and respiratory failure. The reduction in platelet counts correlates with the main blood gas parameters of respiratory failure. Moreover, platelet count values were progressively lower depending on the severity of the clinical scenario and the degree of medical assistance required. In addition, COVID-19 subjects who died, compared to the survivors, had lower platelet values at the time of entry to the emergency room. Therefore, several etiological hypotheses can be argued from our clinical and real-life findings, but further specific studies, both in vitro and in vivo, are required to confirm those data.

In conclusion, a reduction in platelet counts, albeit modest in the early stages of the disease, is one of the typical findings of SARS-CoV-2 infection and correlates with the severity of respiratory failure, representing a prognostic index of increased mortality risk.

## Figures and Tables

**Figure 1 jcm-11-04112-f001:**
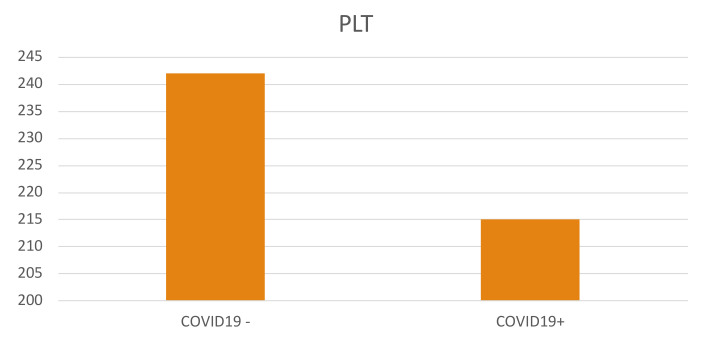
Platelet level in patients without and with SARS-CoV-2 infection at entrance to the Emergency Department. The value indicated for platelets (PLT) is ×10^9^/L. The mean platelet counts were different between the two groups (242.1 ± 92.1 in COVID-19 negative vs. 215.2 ± 82.8 in COVID-19 patients, *p* < 0.001).

**Figure 2 jcm-11-04112-f002:**
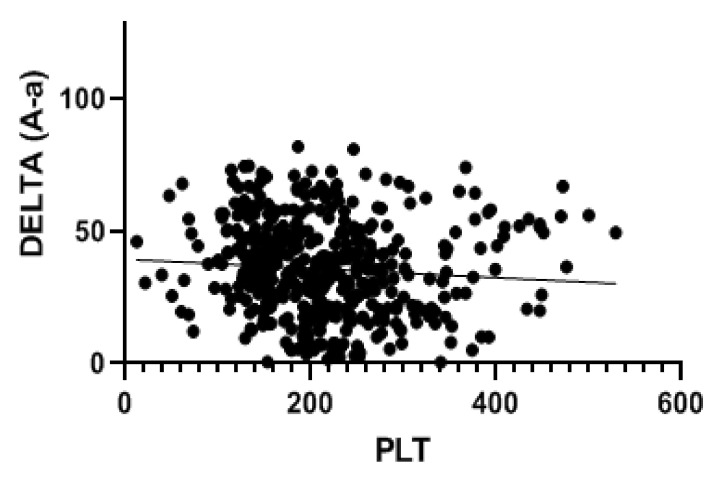
Platelet level in patients with SARS-CoV-2 infection correlated with DELTA (A-a) duringblood gas analysis at entry to the emergency department. The value indicated for platelets (PLT) is ×10^9^/L. In COVID-19 patients, the platelet count correlated with the A-aO_2_ (*p* = 0.001, rho = −0.149).

**Figure 3 jcm-11-04112-f003:**
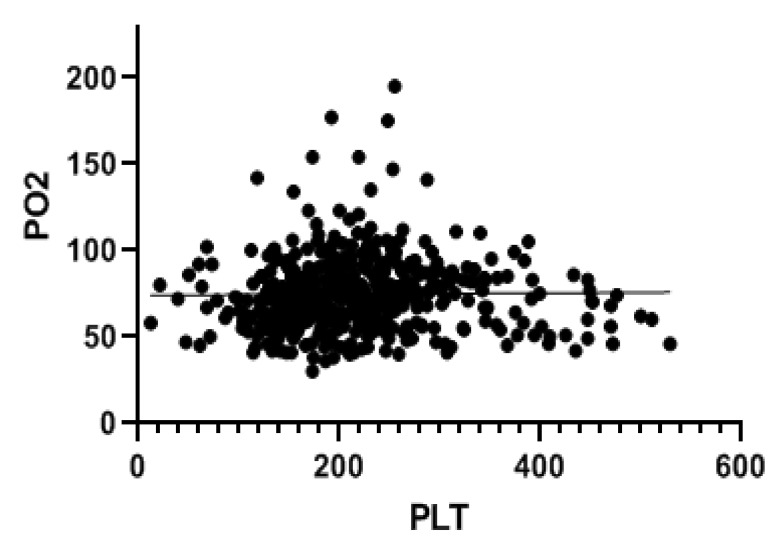
Platelet level in patients with SARS-CoV-2 infection correlated with pO2 during blood gas analysis at entry to the emergency department. The value indicated for platelets (PLT) is ×10^9^/L. In COVID-19 patients, the platelet count correlated with the values of PaO_2_ (*p* = 0.036; rho = 0.093).

**Figure 4 jcm-11-04112-f004:**
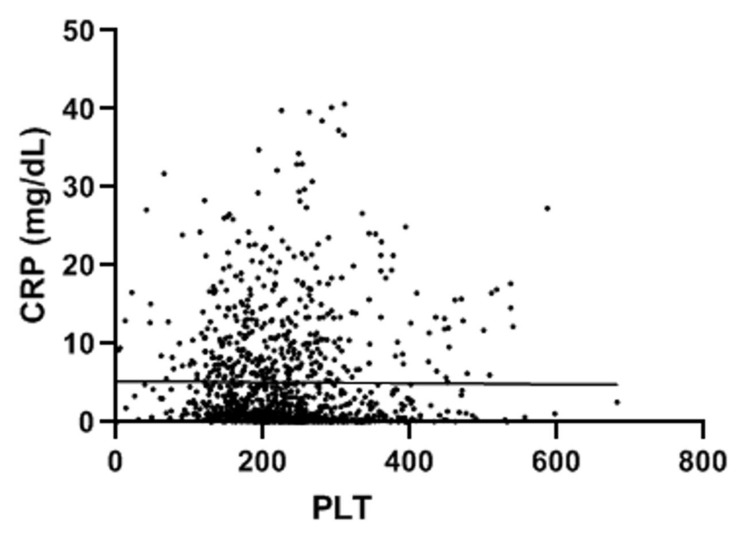
Platelet count inversely correlated with C-reactive protein (*p* < 0.001). The value indicated for platelets (PLT) is ×10^9^/L. CRP: C-reactive protein.

**Figure 5 jcm-11-04112-f005:**
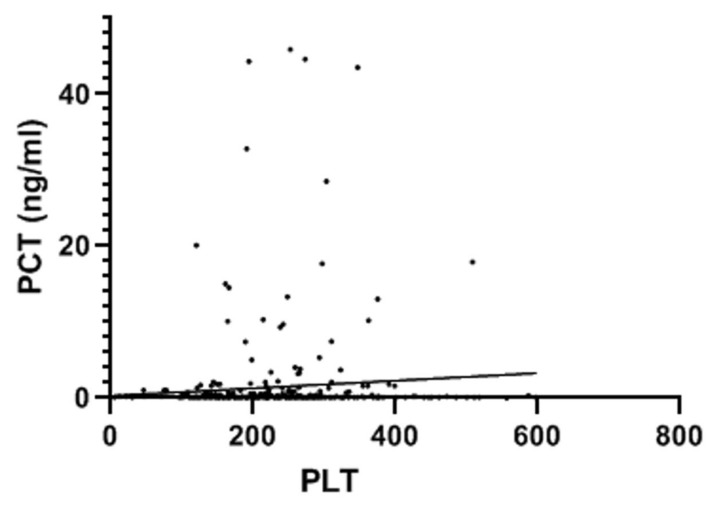
Platelet count did not correlate with procalcitonin (*p* = 0.96). The value indicated for platelets (PLT) × 10^9^/L. PCT: procalcitonin.

**Table 1 jcm-11-04112-t001:** Patients’ characteristics.

	All Patients (998)	COVID-19+ (489)	COVID-19− (509)	*p* Value
Female sex	957 (52.4%)	301 (48.8%)	453 (53.7%)	0.089
Hypertension	561 (30.7%)	238 (38.6)	261 (31%)	<0.001
Diabetes	171 (9.4%)	69 (11.2%)	77 (9.1%)	0.168
COPD	165 (9.0%)	66 (10.7%)	89 (10.5%)	<0.001
Asthma	70 (3.8%)	18 (2.9%)	42 (5%)	0.200
Other lung disease	67 (3.7%)	18 (2.9%)	40 (4.7%)	0.242
Ischaemic cardiac disease	148 (8.1%)	49 (7.9%)	85 (10.1%)	0.002
CKD	122 (6.7%)	48 (7.8%)	62 (7.4%)	0.068
Stroke	90 (4.9%)	36 (5.8%)	45 (5.3%)	0.055

COPD: chronic obstructive pulmonary disease; CKD: chronic kidney disease.

**Table 2 jcm-11-04112-t002:** Patients’ symptoms.

Symptoms	All Patients (998)	COVID-19+ (489)	COVID-19− (509)	*p* Value
Fever	1325 (72.6%)	498 (80.7%)	567 (67.3%)	<0.001
Dyspnea	641 (35.1%)	243 (39.4%)	283 (33.6%)	0.305
Cough	803 (44.0%)	305 (49.4%)	324 (38.4%)	<0.001
Conjunctivitis	42 (2.3%)	8 (1.3%)	27 (3.2%)	0.163
Pharyngodynia	203 (11.1%)	38 (6.2%)	107 (12.7%)	0.000
Headache	225 (12.3%)	55 (8.9%)	99 (11.7%)	<0.001
Asthenia	317 (17.4%)	112 (18.2%)	141 (16.7%)	0.477
Myalgia/arthralgia	219 (12.0%)	76 (12.3%)	98 (11.6%)	0.900
Diarrhea	300 (16.4%)	86 (13.9%)	146 (17.3%)	0.071
Anosmia	73 (4.0%)	30 (4.9%)	25 (3.0%)	0.283
Ageusia	146 (8.0%)	56 (9.1%)	53 (6.3%)	0.033
Chest pain	98 (5.4%)	16 (2.6%)	51 (6.0%)	0.016

**Table 3 jcm-11-04112-t003:** Patients’ vital parameters.

	All Patients (998)	COVID-19+ (489)	COVID-19− (509)	*p* Value
Age (years)	57.0 ± 21.2	62.3 ± 19.3	57.3 ± 21.7	<0.001
SBP (mmHg)	128.3 ± 21.0	125.9 ± 20.2	129.4 ± 22.2	0.032
DBP (mmHg)	75.5 ± 12.8	74.6 ± 12.4	75.6 ± 13.4	0.159
MAP (mmHg)	68.4 ± 42.4	76.7 ± 36.1	68.8 ± 43.2	0.118
HR (bpm)	88.2 ± 16.9	88.9 ± 16.7	89.0 ± 17.1	0.95
RR (a/min)	18.8 ± 5.2	19.7 ± 5.5	18.7 ± 5.2	<0.001
Temperature (°C)	36.9 ± 0.7	37.1 ± 0.8	36.9 ± 0.7	<0.001

SBP: systolic blood pressure; DBP: diastolic blood pressure; MAP: mean arterial pressure; HR: heart rate; RR: respiratory rate.

## Data Availability

The data that support the findings of this study are available from the corresponding author upon request.

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
