# Peer review of "Platelet Count in Patients with SARS-CoV-2 Infection: A Prognostic Factor in COVID-19"

_jcm, 2022, doi:10.3390/jcm11144112_

Round 1

Reviewer 1 Report

The restrospective analysis "Platelet count in patients with SARS-CoV2-2 infection: A Prognostik factor in Covid-19" due to its retrospective nature and  simplified statistical analysis of the patients characteristics inclusive their respective blood gas analysis, lacks novelty according to the prognostic relevance of platelet count. 

Even though the potential pathophysiological mechanisms of Covid-19 induced thrombocytopenia was clearly cited and clearly presented at the discussion part, does not present any additional information about the platelet count in SARS-Covid-19 patients and its relevance on the prognosis of these cohort. There are serious of retrospective analyses and even metaanalyses which described this association after the first wave in 2020. 

A further assessment of adenosine diphosphate (ADP)-, thrombin receptor activator peptide 6 (TRAP)- , and arachidonic acid (AA)-induced platelet activity through impedance aggregometry could be performed to further define the real prognostic impact of thrombocypenia in such a cohort. At the submitted manuscript it is missing. 

Author Response

Thank you for reviewing the work and providing your suggestions. Unfortunately, the retrospective nature of the work and the setting of our specific emergency room did not allow us to perform further specific diagnostic investigations on platelet aggregation. We have included this limitation in the text (lines 189-198) and cited some works that performed this type of evaluation.

Reviewer 2 Report

The manuscript entitled "PLATELET COUNT IN PATIENTS WITH SARS-CoV-2 IN-FECTION: A PROGNOSTIC FACTOR IN COVID-19 "aim to examine platelet count values in patients  infected with SARS-CoV-2 comparing them to a control group consisting of non-COVID-19 patients and trying to explain molecular and physio-pathological hypothesis at the basis of the observed data.  The manuscript was well written and  revealed  reduction in platelet counts correlates with the main blood gas parameters of respiratory failure.  The finding  provided some information to make a hypothesis to explain molecular but could not make a conclusion before more strong evidence. Authors should mention that in the conclusion or limitation.    

Author Response

Thank you for reviewing the work and providing your suggestions. We have included this limitation in the text (lines 206-208).

Round 2

Reviewer 1 Report

The authors of the submitted manuscript correctly replied some the critical points emphasised through the reviewers. However, still some important methodological part should be stressed out in order to make the stated hypothesis based on strong clinical evidence relevant. 

As the further specific diagnostic investigations on platelet aggregation couldn't be performed on this collective, due to its retrospective nature, what about the correlation of the SOFA score for example as e measure for the clinical outcome with markers of platelet, CRP , leukocyte count etc in this civid-positive cohort. This analysis is relevant in this cohort and is missing. 

Author Response

POINT BY POINT RESPONSE

The authors of the submitted manuscript correctly replied some the critical points emphasised through the reviewers. However, still some important methodological part should be stressed out in order to make the stated hypothesis based on strong clinical evidence relevant.

As the further specific diagnostic investigations on platelet aggregation couldn't be performed on this collective, due to its retrospective nature, what about the correlation of the SOFA score for example as e measure for the clinical outcome with markers of platelet, CRP , leukocyte count etc in this civid-positive cohort. This analysis is relevant in this cohort and is missing.

R: Thank you for reviewing our revised paper and providing your suggestions. Regarding your observations on sepsis indexis and inflammatory indexis we evaluated the correlation between CRP and procalcitonin and platelets. Thank to your precious comment, we found that platelets inversely correlated with CRP with statistical significance and this is an expected result in patients with systemic inflammation. On the contrary, platelet value did not correlate with procalcitonin maybe because it represents an index focused on the presence of bacterial sovrainfection that is present only in a part of patients with SARS-CoV-2 infection.

We have included these new data in the text (please see lines 83-86 and lines 176-180) and two new figures (please see figure 4 and figure 5).
